# Pleural Effusion in COVID-19 Pneumonia: Clinical and Prognostic Implications—An Observational, Retrospective Study

**DOI:** 10.3390/jcm12031049

**Published:** 2023-01-29

**Authors:** Sara Cappelli, Elisabetta Casto, Marta Lomi, Alessandra Pagano, Luciano Gabbrielli, Roberta Pancani, Ferruccio Aquilini, Giulia Gemignani, Laura Carrozzi, Alessandro Celi

**Affiliations:** 1Department of Surgical, Medical and Molecular Pathology and Critical Care, University of Pisa, 56126 Pisa, Italy; 2Pneumology Unit, Pisa University Hospital, 56126 Pisa, Italy; 3OU Organization of Hospital Services, Pisa University Hospital, 56126 Pisa, Italy

**Keywords:** pleural effusion, computer tomography, prognosis, COVID-19, total severity score, intensity of care, mortality, length of hospitalization

## Abstract

Background: COVID-19 presents with a wide spectrum of clinical and radiological manifestations, including pleural effusion. The prevalence and prognostic impact of pleural effusion are still not entirely clear. Patients and methods: This is a retrospective, single-center study including a population of consecutive patients admitted to the University Hospital of Cisanello (Pisa) from March 2020 to January 2021 with a positive SARS-CoV-2 nasopharyngeal swab and SARS-CoV-2-related pneumonia. The patients were divided into two populations based on the presence (*n* = 150) or absence (*n* = 515) of pleural effusion on chest CT scan, excluding patients with pre-existing pleural effusion. We collected laboratory data (hemoglobin, leukocytes, platelets, C-reactive protein, procalcitonin), worst PaO_2_/FiO_2_ ratio as an index of respiratory gas exchange impairment, the extent of interstitial involvement related to SARS-CoV-2 pneumonia and data on intensity of care, length of stay and outcome (discharge or death). Results: The prevalence of pleural effusion was 23%. Patients with pleural effusion showed worse gas exchange (*p* < 0.001), longer average hospital stay (*p* < 0.001), need for more health care resources (*p* < 0.001) and higher mortality (*p* < 0.001) compared to patients without pleural effusion. By multivariate analysis, pleural effusion was found to be an independent negative prognostic factor compared with other variables such as increased C-reactive protein, greater extent of pneumonia and older age. Pleural effusion was present at the first CT scan in most patients (68%). Conclusions: Pleural effusion associated with SARS-CoV-2 pneumonia is a relatively frequent finding that is confirmed to be a negative prognostic factor. Identifying early prognostic factors in an endemic-prone disease such as COVID-19 is necessary to optimize its clinical management. Further clinical studies aimed at better characterizing pleural effusion in these patients will be appropriate in order to clarify its pathogenetic role.

## 1. Introduction

COVID-19 is an infectious disease caused by the SARS-CoV-2 coronavirus. This virus was first identified in China in December 2019, and within months spread worldwide until the World Health Organization declared pandemic status in March 2020. To date, nearly 652 million cases have occurred worldwide [1].

SARS-CoV-2 infection has a very wide spectrum of manifestations; it may be asymptomatic, give mild symptoms (cold, fever, pharyngodynia, arthralgias, myalgias, etc.) or pneumonia that may be associated with respiratory failure and require hospitalization. In some cases, an acute respiratory distress syndrome and septic shock to multi-organ failure with high mortality rate may develop [2,3]. The number of deaths worldwide has reached 6.32 million [1].

Computed tomography (CT) of the chest has been widely used to diagnose SARS-CoV-2 pneumonia and to assess its extent and severity [4]. COVID-19 pneumonia presents radiologically with alterations of the lung that are generally bilateral and may affect all lung lobes; these alterations include typical ground-glass interstitial opacities, consolidations, especially in the more advanced stages of the disease and thickening of inter- and intralobular septa giving the parenchyma a crazy paving appearance. Traction bronchiectasis and focal and mainly subpleural architectural distortions may be associated [5,6,7].

Pleural effusion (PE) associated with SARS-CoV-2 pneumonia is less common than the aforementioned abnormalities; however, it has been observed more frequently than in other viral pneumonias [8].

In 2021, a systematic review of 23 studies identified that the average prevalence of pleural effusion in COVID-19 patients is 9.55% and that PE is associated with higher severity of disease [9]. Another review of 47 observational studies identified an extremely wide range, from 0.9% to 100%, of prevalence of PE in the different studies [10].

Indeed, the actual prevalence and possible prognostic role of PE in COVID-19 pneumonia patients are not entirely clear. The purpose of this retrospective study is to contribute to the evaluation of the prevalence of PE in SARS-CoV-2 pneumonia and of its prognostic role.

## 2. Patients and Methods

### 2.1. Study Population

This is a retrospective single-center study which includes all ≥ 18-year-old COVID-19 patients admitted to Cisanello University Hospital (Pisa, Italy) from March 2020 to January 2021.

The starting population included consecutive non-vaccinated patients with SARS-CoV-2 infection verified by nasopharyngeal swab analyzed with reverse transcriptase-polymerase chain reaction. All patients were found to be SARS-CoV-2 positive upon admission to the emergency room. Exclusion criteria were missing critical information, chest CT scan not performed on admission to the Emergency Department, absence of pneumonia at chest CT scan at admission, preexisting conditions that prevented the correct evaluation of pneumonia and evidence of PE in a previous chest CT scan available in our archives.

Patients were divided into two distinct populations depending on whether PE was present. Although all images were reviewed by the authors (see below), only effusions described by the radiologist in the official report were considered. Accordingly, the 15 patients in which a small (<1 cm) effusion observed by us but not reported by the radiologist were considered PE −. The PE was quantified by measuring its thickness on CT scans. We recorded whether it was unilateral (left or right) or bilateral and whether it was greater or equal to 1 cm or less than 1 cm. We also collected the following data: laboratory data on admission to the Emergency Department including hemoglobin, white blood cells, platelets and C-reactive protein (CRP); procalcitonin (PCT); worst PaO_2_/FiO_2_ ratio during hospitalization (P/F nadir); intensity of care (see below); length of stay; and outcome (expressed as discharge or death) were also recorded. Intensity of care refers to the type of treatment received by the patients: medical therapy only, oxygen therapy, non-invasive ventilation/high-flow nasal cannula (HFNC) or invasive ventilation. Interstitial engagement related to SARS-CoV-2 pneumonia was assessed by calculating the extent of changes (e.g., ground-glass opacities, lung thickening, crazy paving) in the five lobes using the total severity score (TSS), which was scored as: <25%, 25–50%, 51–75% or >75% [11]. Each chest CT scan was evaluated by at least two physicians within the same session; in the unlikely event of disagreement (<10%) a third reader was consulted until an agreement was obtained.

Since all the procedures described in this study were part of routine care for COVID-19 patients, this study was conducted in compliance with the Declaration of Helsinki and with the approval of the local Ethic Committee (protocol CEAVNO 2020-17241), but patients' informed consent was waived. 

### 2.2. Statistical Analysis

Continuous variables are shown as mean ± SD or median [interquartile range] as appropriate. Categorical variables are shown as counts and percentages. Independent *t*-tests were used to compare the mean of continuous variables when the data had a normal distribution according with the results of the Shapiro test; when the distribution was not normal, the Mann–Whitney U test was used. Categorical variables were compared by the χ² test. Survival analysis was performed using the Kaplan–Meier method followed by log-rank analysis. We used multivariate logistic regression to evaluate the role of PE, as well as other factors that might have affected survival. A two-tail *p* value <0.05 was considered statistically significant. Jamovi [12], SPSS version 28.0.1 for Windows (IBM corporation, Armonk, NY, USA) and GraphPad Prism 5 (GraphPad software, San Diego, CA, USA) software were used to perform statistical analyses and prepare the graphs.

## 3. Results

During the study period, 1104 patients were admitted with a positive nasopharyngeal swab for SARS-CoV-2. Of these, 243 were excluded due to missing information, 139 patients were excluded due to the absence of chest CT scan on admission to the Emergency Department, 40 patients were excluded due to the absence of pneumonia at chest CT scan and 1 patient was excluded because a pre-existent lymphangioleiomyomatosis on chest CT scan prevented the correct evaluation of pneumonia.

The population included in this study comprised 681 subjects (432 males) older than 18 years. They were divided into patients with (*n* = 166; PE +) and without (*n* = 515; PE−. A total of 16 subjects presenting with PE prior to admission were excluded from the case group. Thus, the final case population was 150 subjects, including 101 males and 49 females. 

Table 1 describes the anthropometric and clinical characteristics of the patients. Of the 515 controls, 320 were male. PE + patients were older (75 ± 13 vs. 67 ± 15 years), had slightly lower hemoglobin levels and had significantly higher C-reactive protein levels. 

PE was unilateral in 35 patients and bilateral in 115. Of the 35 unilateral effusions, 18 were ≥1 cm and 17 were < 1 cm. Of the 115 bilateral effusions, 53 were ≥1 cm bilaterally, 35 were <1 cm bilaterally and 27 were ≥1 cm unilaterally (Table 2). The median number of days from admission to the Emergency Department until the first observation of PE was 1 [IQR 5; minimum 0–maximum 33].

When the level of gas exchange impairment was analyzed in the two groups of patients, PE + patients showed a significantly lower P/F nadir (167 ± 9 mmHg vs 235 ± 5 mmHg; *p* < 0.001) (Figure 1A). Figure 1B shows the difference in the length of stay in the two groups; the average number of hospitalization days was 23.3 ± 1.5 and 14.9 ± 0.4 in PE + and PE − patients, respectively (*p* < 0.001)

When evaluating pneumonia severity between the two groups by using the TSS on the CT scan, there was a trend towards a higher parenchymal engagement in PE + patients, even though it did not reach significance (*p* = 0.054) (Figure 2).

Figure 3 compares the level of care in the two groups. The difference between the two groups was statistically significant (*p* < 0.001), with PE + patients requiring higher health resources (namely, mechanical ventilation, either invasive or not). 

In-hospital mortality rate was 36.7% in PE + patients, while it was only 13.8% in PE − group; this difference was statistically significant (*p* < 0.001, not shown). 

Figure 4 shows the results obtained from the survival analysis with the Kaplan–Meier curve in the two groups. The survival fraction was significantly lower in the PE + group.

We performed a binomial logistic regression, including the variables age, presence of effusion, TSS at admission and CRP at admission (Table 3). The probability of death increases by 149% in the presence of PE.

## 4. Discussion

Our data indicate a prevalence of new-onset PE in a large series of consecutive patients referred to a third level university hospital, 23% of which had a CT scan, widely considered the gold standard for the diagnosis for PE, available. The reported prevalence of PE in COVID-19 is extremely variable. However, some studies describe a relatively small series [13,14], or refer to patients belonging to special populations [15] or are based on standard chest X-ray or lung ultrasound [16,17,18].

A study in a Chinese population with apparently similar inclusion criteria reported a much smaller prevalence (9.2%); the population of this study was, however, significantly younger (51 ± 16 years) [19]. Finally, a multicenter study in an Italian geriatric cohort with a mean age of 78 showed prevalence data comparable to ours (23%) [20].

Most available studies do not attempt to quantify the effusion; therefore, a comparison with previous data under that respect is not feasible. As for the localization of PE, this was bilateral in most cases, in agreement with a previous reports [20]. However, there seems not to be a significant prognostic implication in the monolateral or bilateral pleural involvement.

The presence of PE correlates with worse clinical outcomes, including gas exchange impairment and increased length of stay; furthermore, PE + patients were more likely to require mechanical ventilation, either invasive or non-invasive. This observation is in agreement with most previously available data.

Finally, survival analysis shows a statistically significant lower survival between PE + and PE − patients, confirming a negative prognostic value for PE. 

PE was associated with more severe interstitial involvement as assessed by TSS, although the association missed the prespecified level of significance. However, multivariate analysis confirmed that PE was an independent predictor of mortality. Both parameters significantly contributed to the prognostic stratification. Age and CRP also proved to be independent predictors of prognosis. 

The main limit of this study is the relatively poor characterization of the patients. The presence of comorbidities (e.g., cancer, renal or heart failure, etc.) have probably impacted the patients’ outcome. However, due to the overwhelming workload during the COVID-19 pandemic that made it extremely difficult to properly record clinical data, such information is not available. The retrospective design is another limit of this study. Finally, this study is monocentric, which might limit its generalizability. On the other hand, this latter characteristic makes the data particularly reliable.

In conclusion, our study shows that PE, more frequently bilateral, is relatively frequent in COVID-19-associated pneumonia and confirms that it represents an independent prognostic factor.

The identification of negative prognostic factors in a disease as prevalent as SARS-CoV-2-related pneumonia, a disease that is likely becoming endemic, is necessary for the optimal management of these patients and to define the most appropriate treatment setting. Of note, PE appeared quite early in our patients, being present at the admission CT scan in 102 (68%) patients; this makes the assessment of the presence of PE potentially useful in the first line assessment of the disease. The widespread diffusion of effective vaccines and, possibly, the emergence of new viral strains have changed the clinical scenario of COVID-19, even though severe pneumonias are still relatively frequent in specific subsets of patients (e.g., immunocompromised patients or individuals who decided to not undergo vaccination). We do not know whether the data presented in this study will be generalizable to this new situation. Further studies to better characterize the effusion (e.g., exudate vs. transudate) will also be helpful in order to investigate the pathogenetic significance of our observations. 

## Figures and Tables

**Figure 1 jcm-12-01049-f001:**
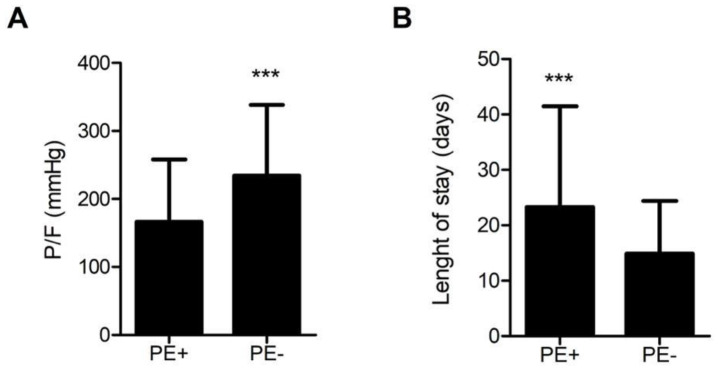
PaO_2_/FiO_2_ (P/F) for PE + and PE − patients. *** *p* < 0.001 by Student's *t*-test (**A**); length of stay (days) for PE + and PE − patients. *** *p* < 0.001 by Student’s *t*-test (**B**).

**Figure 2 jcm-12-01049-f002:**
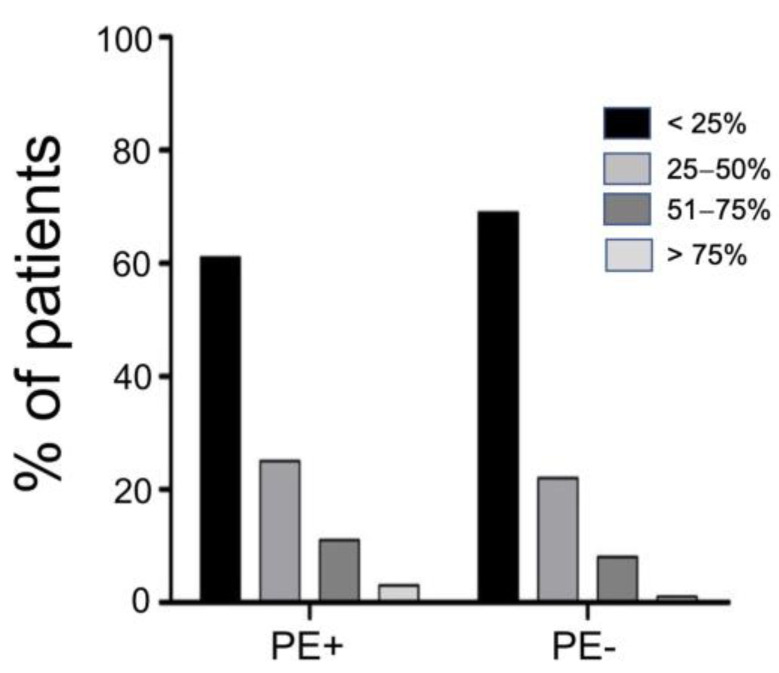
Total severity score (TSS) for PE + and PE − patients. *p* = 0.054 by χ² test.

**Figure 3 jcm-12-01049-f003:**
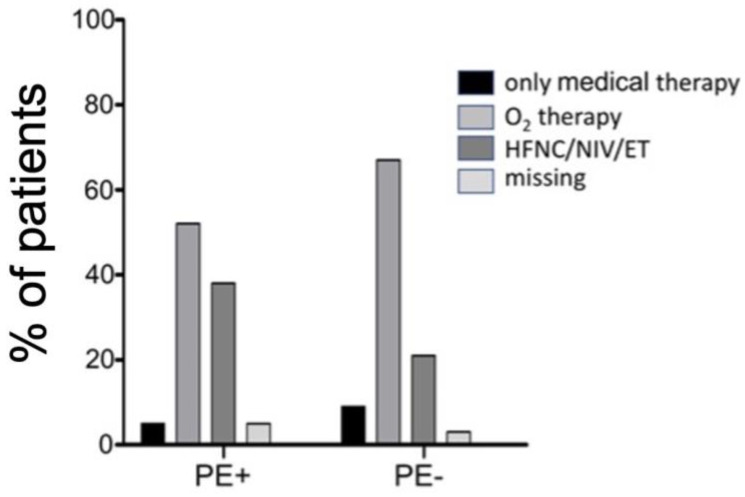
Level of care (HFNC: high-flow nasal cannula; CPAP: continuous positive airway pressure; NIV: non-invasive ventilation; ET: endotracheal intubation) for PE + and PE − patients. *p* < 0.001 by χ² test.

**Figure 4 jcm-12-01049-f004:**
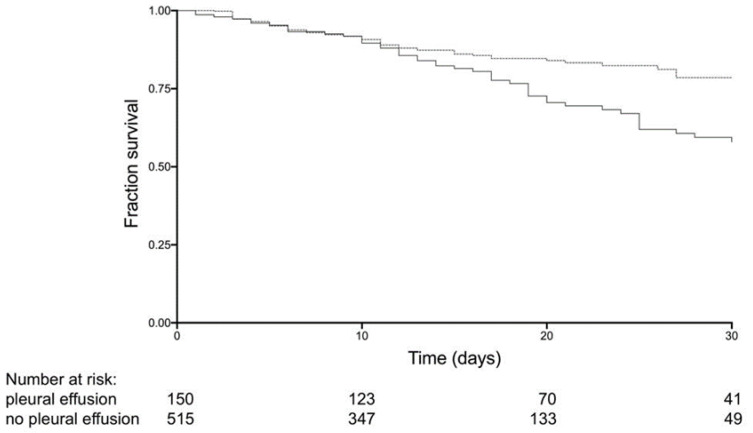
Kaplan–Meier curve for the probability of survival for PE + (solid line) and PE − (dashed line).

**Table 1 jcm-12-01049-t001:** Characteristics of the patients.

	PE + (*n* = 150)	PE − (*n* = 515)	*p* Value
Age, years (mean ± SD)	75.0 ± 13.2	67.0 ± 15.2	<0.001
Male sex	101 (67.3)	320 (62.1)	=0.24
Blood tests			
Hb (g/dL) (mean ± SD)	12.5 ± 2.3	13.5 ± 2.1	<0.001
WBC 10³/µL (mean ± SD)	9208 ± 6082	8187 ± 7454	=0.13
PLT 10³/µL (mean ± SD)	205,246 ± 92,417	209,959 ± 101,903	=0.61
CRP mg/dL median [IQR]	7.43 [8.89]	5.54 [8.76]	<0.01

Hb: hemoglobin; WBC: white blood cells; PLT: platelets; CRP: C-reactive protein; SD: standard deviation; IQR: interquartile range.

**Table 2 jcm-12-01049-t002:** Characteristics of PE.

	Unilateral (*n* = 35)		
	Left	Right	Tot
<1 cm	7	10	17
≥1 cm	13	5	18
		Bilateral (*n* = 115)	
<1 cm bilaterally		35	
≥1 cm bilaterally		53	
≥1 cm monolaterally		27	

**Table 3 jcm-12-01049-t003:** Multivariate logistic analysis for in hospital mortality.

Variable	OR [95% CI]	*p*
Age (years)	1.08 [1.05–1.10]	<0.001
PE no (reference) PE yes	1 2.49 [1.54–4.03]	<0.001
TSS <25% (reference) 25–50% 51–75% >75%	1 1.70 [1.01–2.86] 3.42 [1.67–6.97] 49.30 [5.07–479.82]	<0.044 <0.001 <0.001
CRP (mg/mL)	1.05 [1.01–1.08]	<0.001

TSS: total severity score; CRP: C-reactive protein; OR: odds ratio; CI: confidence interval.

## Data Availability

The data presented in this study are available on request from the corresponding author. The data are not publicly available due to [insert reason here].

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
