# Peer review of "Pleural Effusion in COVID-19 Pneumonia: Clinical and Prognostic Implications—An Observational, Retrospective Study"

_jcm, 2023, doi:10.3390/jcm12031049_

Round 1

Reviewer 1 Report

As a clinician treating patients with covid-19 since March 2020, I am aware of the importance of prognostic factors in this disease.

The authors evaluate a retrospective but large series of patients with COVID-19, describing the prevalence of pleural effusion and its prognostic implications. They observe that it has a prevalence of 23%, with a longer hospital stay and higher mortality.

The abstract is concise and clear, containing all the important data and conclusions.

The introduction is up-to-date and situates the study in the current state of knowledge on the subject.

The population and study period are well defined in Patients and methods. The statistical methodology is impeccable. However, the registration number of the report of the Ethics Committee that approved the study should be cited.

In the results section, it would be interesting if the researchers had more information on the patients, such as the days of the clinic until the diagnosis of COVID-19 or pleural effusion, and some comorbidities of interest, such as heart disease. It would also be interesting to know the severity on admission (with general scores such as SOFA or APACHE II, or specific scores for pneumonia, such as Fine, CURB-65, or even for viral pneumonia, such as MuLBSTA).

The discussion is well planned and compares the results with the existing bibliography.

The study was carried out at the beginning of the pandemic, with unvaccinated patients. Do the authors consider that with the change in clinical manifestations after vaccination campaigns and the different strains of the virus, the conclusions of the study will be generalizable to the current situation?

The conclusion is clear and is supported by the results of the study.

Minor changes:
1. Page 1 line 39: reference 1 has not date of consult, "To date" needs to be clarify.

2. The title of table 3 should be more extensive, specifying what the multivariate analysis is performed on. For the context, it is about mortality, but the title of the table should stand alone.

Author Response

First, we would like to thank the reviewer for his/her comments, that we have tried to address in the following point-by-point reply.

"However, the registration number of the report of the Ethics Committee that approved the study should be cited."

The number of the Ethic Committee approval has been added.

"it would be interesting if the researchers had more information on the patients, such as the days of the clinic until the diagnosis of COVID-19 or pleural effusion"

All patients were diagnosed upon arrival to the the emergency Department. We added a sentence to better clarify this. Pleural effusion was generally recognised at the baseline CT scan. However, the range of this figure is reported in the manuscript.

"(...) some comorbidities of interest, such as heart disease.

The issue of comorbidities is indeed critical. Unfortunately, owing to the overwhelming workload during the first two waves of COVID-19, only relatively few clinical data were recorded upon admission. We stressed this obvious limit of the study in the revised version of the manuscript.

"It would also be interesting to know the severity on admission (with general scores such as SOFA or APACHE II, or specific scores for pneumonia, such as Fine, CURB-65, or even for viral pneumonia, such as MuLBSTA)."

The same considerations reported above apply to the lack of the necessary data to calculate the different severity scores (e.g. bacterial superinfection, daily urine output, etc.). We did obtain, and report, the Total Severity Score from the CT scans. This semiquantitative score has been proven a reliable approach to estimate disease severity (see, for example, Inoue et al., Comparison of semiquantitative chest CT scoring systems to estimate severity in coronavirus disease 2019 (COVID19) pneumonia; Eur Radiol, (2022) 32:3513–3524). We added a sentence in the manuscript to acknowledge this issue.

"Do the authors consider that with the change in clinical manifestations after vaccination campaigns and the different strains of the virus, the conclusions of the study will be generalizable to the current situation?"

The clinical scenario of COVID-19 has indeed changed over the last several months. It is likely that our data will not be generalisable. One might argue, however, that while vaccinations have reduced the incidence of severe pneumonias, such cases still exist (for example in immunocompromised patients or in individual who decided to not undergo vaccination) and they might have a similar clinical course than the original unvaccinated population. Of course this is just a speculation and new data will be necessary.  We added a sentence to the manuscript to acknowledge this issue. 

"Page 1 line 39: reference 1 has not date of consult, "To date" needs to be clarify."

We updated the number and reported the date of consult.

"The title of table 3 should be more extensive, specifying what the multivariate analysis is performed on. For the context, it is about mortality, but the title of the table should stand alone."

We modified the title of table 3.

Reviewer 2 Report

The study is interesting and well designed. However, there is a lack of data on population characteristics in Table 1 (no information on comorbidities in patients between PE+ and PE-, in particular, information on concomitant heart failure, cancer and/or other diseases that may affect clinical outcome should be added). Another problem is the lack of information on COVID-19 complications during hospitalization (e.g., DIC, cardiac and neurological complications, renal failure ). Finally, I don't see p-values for different degrees of COVID-19 severity between PE+ and PE- in Figure 3. The discussion section should include possible clinical factors which may affect the outcome in these patients.

Author Response

First, we would like to thank the reviewer for her/his comments, which we tried to address in the following point-by-point reply.

"(...) there is a lack of data on population characteristics in Table 1 (no information on comorbidities in patients between PE+ and PE-, in particular, information on concomitant heart failure, cancer and/or other diseases that may affect clinical outcome should be added). Another problem is the lack of information on COVID-19 complications during hospitalization (e.g., DIC, cardiac and neurological complications, renal failure ). Another problem is the lack of information on COVID-19 complications during hospitalization (e.g., DIC, cardiac and neurological complications, renal failure 

The lack of information concerning comorbidities and, in general, the relatively inadequate clinical characterisation of the patients is a clear limit of this manuscript. Unfortunately, owing to the overwhelming workload during the first two waves of COVID-19, only relatively few clinical data were recorded. We stressed this obvious limit of the study in the revised version of the manuscript, but we are not able to retrieve the necessary information at this point. We could also argue that the main aim of this study was to investigate the prognostic role of pleural effusion, independent of its pathogenesis. In other words, if we satisfactorily demonstrate that a patient's prognosis is influenced by the presence of a pleural effusion, this observation remains true whether the effusion is secondary to a preexisting comorbidity (e.g renal insufficiency or cardiac failure) or to the COVID-19 related pneumonia.  

"I don't see p-values for different degrees of COVID-19 severity between PE+ and PE- in Figure 3"

We added the p-value in the legend of figure 3.

"The discussion section should include possible clinical factors which may affect the outcome in these patients."

We modified the discussion to account for this observation.

Round 2

Reviewer 2 Report

Thank you for responses and added limitations.